# Addressing the ethical issues raised by synthetic human entities with embryo-like features

**Abstract** The "14-day rule" for embryo research stipulates that experiments with intact human embryos must not allow them to develop beyond 14 days or the appearance of the primitive streak. However, recent experiments showing that suitably cultured human pluripotent stem cells can self-organize and recapitulate embryonic features have highlighted difficulties with the 14-day rule and led to calls for its reassessment. Here we argue that these and related experiments raise more foundational issues that cannot be fixed by adjusting the 14-day rule, because the framework underlying the rule cannot adequately describe the ways by which synthetic human entities with embryo-like features (SHEEFs) might develop morally concerning features through altered forms of development. We propose that limits on research with SHEEFs be based as directly as possible on the generation of such features, and recommend that the research and bioethics communities lead a wide-ranging inquiry aimed at mapping out solutions to the ethical problems raised by them.

**JOHN AACH**[*], **JEANTINE LUNSHOF, ESWAR IYER AND GEORGE M  CHURCH**[*]

**\*For correspondence:** aach@ genetics.med.harvard.edu (JA); gchurch@genetics.med.harvard. edu (GMC)

Human pluripotent stem cells (hPSCs) are able to generate whole organisms through cloning, to generate cell types from all three germ layers in culture, and to generate organoids. However, hPSCs have not been recognized as totipotent: that is, they have been thought to be incapable of self-organizing and executing a body plan on their own (*Denker, 2014*). Experiments with embryoid bodies generated from isolated pluripotent stem cells (PSC) in culture, conducted first with mouse cells (*Bedzhov and Zernicka-Goetz, 2014*) and later with human cells (*Deglincerti et al., 2016*; *Shahbazi et al., 2016*; see also *Itskovitz-Eldor et al., 2000*), have found that these bodies can recapitulate early epiblastic polarization and lumen formation in a way that is similar to the early stages of pro-amnionic cavity development in actual embryos, but they do not develop later epiblastic features. However, Warmflash et al. reported that hPSC that had been micropatterned into colonies of controlled size and shape developed features identifiable as a primitive streak and ordered layers of cells from all three germ layers after treatment with BMP4 (*Warmflash et al., 2014*).

These features and structures did not emerge in the absence of micropatterning. Commentaries (*Hyun et al., 2016*; *Pera et al., 2015*) have noted that these experiments represent an important advance by showing how hPSCs can be used to model early human development by dint of a previously unsuspected capacity for totipotency. However, they also raise complex issues regarding how embryo research is regulated, including with the 14-day rule.

The 14-day rule arose from recommendations by a series of commissions dating back to 1979 (*Warnock, 1984*; *National Institutes of Health, 1994*; *Ethics Advisory Board, 1979*) charged with working out ethics-based guidelines for embryo creation and usage, initially focused on assisted reproductive technologies but later extended to stem cells and other biological constructs. These guidelines covered issues ranging from the provenance, procurement, and handling of embryos and germ cells, their usage in assisted reproductive technologies, and permissions and informed consents, but also increasingly specified limits on the types of experimental research that could be conducted

with embryos, stem cells and other constructs. Here we call these latter "research limits" to distinguish them from these other kinds of guidelines, and note that we not only focus exclusively on such limits to experimental research *vs.* these other guidelines, but also on the ethical and scientific reasons for establishing these limits *vs.* the ways in which they may be variously rendered as laws, policies, or other regulations.

The research limit specified by the 14-day rule is that experiments with intact human embryos must not allow them to develop beyond the earlier of day 14 or the appearance of the primitive streak (PS). In formulating and maintaining the rule, each commission in turn considered and came to a collective judgment that early embryos, while entitled to special respect and protection, could yet be subjects of well-justified but stringently regulated experimental research (see *Ethics Advisory Board, 1979*, pp. 100-1; *Warnock, 1984*, pp.62-4; *National Institutes of Health, 1994*, chap. 3, p. 50). Nevertheless the rule was never unanimously accepted, and dissents and reservations were recorded both within and outside of the adopting commissions (*New Scientist, 1984*; *Nature, 1989*; *National Institutes of Health, 1994*; *Warnock, 1984*, sections 11.11-14, pp. 61-2; Expression of Dissent: B, p. 90-3). These divisions were mainly anchored in conflicting and deeply-held opinions concerning the moral status of embryos – especially between the view of some well-established traditions that embryos have full moral status from the moment of conception, and a less cohesive but yet widely-held set of views that see embryos as increasing in moral status as they develop (*National Institutes of Health, 1994*, chap. 3, p. 40). We describe these latter views as agreeing with the thesis that *moral status is developmentally emergent*. Moreover, the specific stipulations of 14 days and the PS were adopted not because they were recognized as having intrinsic moral significance, but rather because they preceded the appearance of more morally significant features and provided unambiguous policy criteria for directing when to terminate experiments (*Warnock, 1984*, sections 11.19–11.22; *National Institutes of Health, 1994*, chap. 3, pp. 46-8). The 14-day rule has thus always been at best an uneasy compromise between divergent moral views, complex biology, and the needs of policy. Nevertheless, it has been viewed as "tremendously successful" because, despite its deficiencies, it has provided a practical mechanism for allowing important scientific

research on embryos to proceed (*Hyun et al., 2016*).

The entities generated by Warmflash et al. (called "gastruloids" by Pera et al., a term used earlier by *van den Brink et al., 2014*) exhibit a PS but they do not violate the 14-day rule because they are clearly not intact embryos: Indeed, part of their promise lies precisely in the circumstance that aspects of human embryo development can be analyzed experimentally in gastruloids in ways that would be considered impermissible with actual embryos by dint of the rule. But the commentaries foresee that these gastruloids can be made more embryo-like and that, given sufficient scientific progress, they could eventually recapitulate development well enough to be considered "synthetic embryos" (*Denker, 2014*) or "embryos in a dish" (*Pera et al., 2015*) that might need to be regulated under the rule. (In view of the possibility of "synthetic embryos", we will call embryos formed from zygotes in culture or through sexual intercourse or assisted insemination (*Cantineau et al., 2013*; *Hurd et al., 1993*) "non-synthetic embryos" where "embryo" alone is ambiguous.) Indeed, in its recently revised guidelines for stem cell research, the International Society For Stem Cell Research (ISSCR) has promulgated the first formal guidelines recognizing this issue by recommending that experiments with "embryo-like structures that might manifest human organismal potential" be reviewed by a proposed human Embryo Research Oversight (EMRO) process, and prohibited if they violate the 14-day rule (see *International Society for Stem Cell Research, 2016*, esp. Recommendation 2.1.3; *Daley et al., 2016*; *Kimmelman et al., 2016*). Continued progress in modeling embryonic development thus poses a dilemma in that the illuminating window that gastruloids open on development could be closed precisely as they become better and more accurate. But the commentaries also note that attempts to apply such a regulation will be encumbered by a circularity because experiments on non-synthetic embryos at the PS stage would be needed to assess how well they correspond to those of gastruloids or embryo-like structures, and these would be prohibited by the 14-day rule. Several commentators (*Hyun et al., 2016*; *Pera et al., 2015*) thus suggest that the 14-day rule itself should be reviewed and possibly adjusted to permit more extended research, and recent experiments (*Deglincerti et al., 2016*; *Shahbazi et al., 2016*) using new methods enabling human embryos to

be cultured up to day 13 have provided additional impetus to this call by casting a spotlight on how little is known about human *vs.* model animal embryo development (*Hyun et al., 2016*). Finally, especially as argued in (*Pera et al., 2015*), gastruloids and synthetic embryos will increasingly showcase the unsustainability of the 14-day rule because differing definitions of "embryo" in existing ethical and legal implementations of the rule predict inconsistent and uncertain handling of these entities across different jurisdictions. These pressures to review the rule reflect an unleashing of tensions inherent in it since its inception by scientific advances that are only now pushing up against the boundaries it specified.

Here we argue that these advances actually pose a deeper challenge to current ethics-based embryo and stem-cell research guidelines that cannot be met by revisiting and adjusting the 14-day rule. We identify the larger difficulties with the guidelines and suggest how these may be overcome, and call for the scientific and bioethics research communities to lead a broad-based, multidisciplinary effort to discuss, critically analyze, and map out solutions to these issues. Put briefly, we claim that new techniques in synthetic biology are enabling the engineering of complex and organized human tissue assemblies, including, now, some that present embryonic features and have the potential to continue to develop additional and more mature features. Here we call these *Synthetic Human Entities with Embryo-like Features (SHEEFs)*. The 'gastruloids' of *Warmflash et al. (2014)* are a simple form of SHEEF but rapid advances in technology will soon enable generation of many new varieties. Through their enhanced engineerability, SHEEFs offer a possible way of escaping the dilemma above by enabling generation of human entities that recapitulate aspects of embryonic development potentially very precisely, but that are different enough from non-synthetic embryos to justify their exemption from research limits on such embryos. But to achieve this will require deep consideration of the conditions under which SHEEFs might develop features that are morally concerning, and a framework that allows research limits to be specified for them.

Below we elaborate specific aspects of this argument: In section 1 we describe the confluence of synthetic biology methods that are giving rise to SHEEFs. In section 2 we show how these SHEEFs raise moral concerns that cannot be addressed by adjusting the 14-day rule because this rule refers research limits to the standard sequence of stages understood to comprise normal embryonic development (hereby called *canonical embryogenesis*), while the new methods that give rise to SHEEFs may enable such stages to be bypassed. In section 3 we describe how current guidelines use canonical embryogenesis as an underlying conceptual model for research limits generally, and how this also leads to a structural difficulty that complicates their extension to SHEEFs. In section 4 we present our core proposals: We argue that these difficulties can be overcome only by basing research limits as directly as possible on the development of features that directly trigger moral concern *vs.* pre-emptive limits based in canonical embryogenesis such as the 14-day rule, and call for a wide-ranging set of exploratory inquiries led by the research and bioethics communities into the ethical and conceptual issues raised by such an undertaking. We also provide high-level suggestions for how a framework for specifying such limits could be formulated and for how these inquiries might be organized. In section 5 we identify additional issues that could be considered by these inquiries, including how it might interface with current interests in revisiting the 14-day rule, impacts of SHEEFs on non-synthetic embryo research, and questions of definition and boundary. In section 6 we suggest how our proposed exploratory inquiries might be initiated.

# 1 Synthetic biology techniques are enabling generation of Synthetic Human Entities with Embryo-like Features (SHEEFs)

The ability to generate SHEEFs is emerging from a confluence of advances in stem cell and tissue engineering and organoid development, which we broadly denote as synthetic biology methods. Since human Embryonic Stem Cells (hESC) were first derived in 1998 (*Thomson et al., 1998*), and human induced Pluripotent Stem Cells (hiPSC) in 2007 (*Takahashi et al., 2007*), researchers have worked to deliver on their promise for generating functional human tissues for therapy and research. In part due to ethical concerns with and regulations on human cloning (*Pattinson and Caulfield, 2004*), much of this effort has been channeled into development of in vitro methods for producing these tissues. The usual starting point has been to identify media factor combinations that induce hPSC differentiation towards a target cell fate and suppress differentiation to others. However, such

regimes only approximate natural development and frequently result in cell types and tissues that are immature, incomplete, and poorly organized compared to mature forms. Recently, 3D tissue culturing methods have been found to yield more complex and realistic organ models (organoids) that feature multiple cell types and patterning characteristic of mature organs (*Lancaster and Knoblich, 2014*). But organoids still present significant limitations: Media factors affect the entire culture and so complicate the creation of organoids whose native tissues derive from different germ layers and divergent lineages. Organoid development and growth are also frequently stunted due to lack of effective means for delivering nutrients to and eliminating wastes from cells in the organoid interior (see, e.g., *Lancaster et al., 2013*). But advances in synthetic biology methods that can outfit hPSC with inducible developmental pathways, and new tissue engineering methods that confer control over the spatial organization of tissue assemblages, are providing approaches for overcoming these limitations. Because driving hPSC differentiation by induced overexpression of transcription factors can powerfully override media signals (e.g., neurons can be derived from hPSC in 4 days even in pluripotency-maintaining media; *Busskamp et al., 2014*), mixed colonies of engineered hPSC can be created that differentiate into divergent lineages, and inducers can also be scheduled to control the timing and order of appearance of these cell types. Meanwhile, cells of different types can be positioned with precision using 3D tissue printing (*Homan et al., 2016*), and microfabrication methods (frequently used in Organs-on-Chips approaches; *Huh et al., 2012*; *Wang et al., 2014*) can also be used to create substrates that facilitate cell type development, adhesion, and tissue organization. 3D printing can also be used to print vasculature into 3D cell cultures and this vasculature can be connected to external perfusion apparatus (*Kolesky et al., 2014*). The convergence of these hPSC genetic and tissue engineering methods is also enabling tissue and organ models to be increasingly derived exclusively from hPSC *vs.* mixtures of primary and tumor cell lines used in earlier artificial tissues, leading at the same time both to models that are more realistic, and to models developed from a common hPSC ground state. These advances are setting the stage for complex and structured SHEEFs that can recapitulate aspects of embryogenesis, and have the potential to both progressively develop into more mature forms as they also accumulate new embryonic features.

While these capabilities are only now emerging, they are advancing rapidly and new methods that offer distinct new capabilities are in development behind them, e.g., the ability to program cells or cell-containing droplets to specifically interact with each other or other substrates using DNA barcodes (*Mali et al., 2013*; *Qi et al., 2013*). Importantly, however, even the relatively simple gastruloid SHEEFs generated in the *Warmflash et al. (2014)* experiments *via* hPSC colony micropatterning gave rise not only to a well-formed embryonic feature (the PS) never previously generated in vitro from hPSC, but to a PS whose circular shape accommodated the shape of the micropattern and differed markedly from the linear PS of non-synthetic embryos. This observation suggests that the developmental potential harbored by hPSC can adapt readily to non-natural conditions and unfold with considerable plasticity, and so forecasts the likelihood that while these advancing methods can be expected to enable SHEEFs to be generated to be increasingly like non-synthetic embryos (as predicted by the commentaries above), we should also expect many kinds of SHEEFs that will be very unlike such embryos.

## 2 SHEEFs raise ethical issues that cannot be addressed by adjusting the 14-day rule because they can bypass canonical embryological stages

A key concern of the commissions that adopted the 14-day rule was to prevent the possibility of embryos experiencing pain or sentience in the context of an experiment. The particular choice of the PS and day 14 as research limits was based on the biological understanding that neurulation proceeds in embryos immediately after the appearance of the PS, so that stopping experiments at the PS would forestall the formation of a nervous system and brain that could, with further development, be subject to these experiences. Thus, while the 1984 Warnock committee report noted "a wide range of opinion" on the question of how long embryos used in research should be kept alive, its leading statement on this point was that, according to a "strictly utilitarian view . . . the ethics of experiments on embryos must be determined by the balance of benefit over harm, or pleasure over pain. Therefore, as long as the embryo is incapable of feeling pain, it is argued that its treatment

does not weigh in the balance." Noting that this still left considerable leeway regarding when the time limit should be set between the beginnings of neurulation and functional neural activity, they conservatively choose the former and "[subtracted] a few days in order that there would be no possibility of the embryo feeling pain," ultimately deciding on the PS (*Warnock, 1984*, section 11.20, p. 65 and section 11.22, p. 66). Similarly, the 1994 NIH Report of the Human Embryo Research Panel (hereby denoted RHERP), likewise considered many options but ultimately focused on "sentience and the ability to experience pleasure and pain" (*National Institutes of Health, 1994*, chap. 3, p. 47; see also p. 37), and again accepted the appearance of the PS as a research limit because it closely precedes neurulation, noting that "[t]here is no neural tissue whatsoever prior to the primitive streak; hence there is no possibility of any kind of sentience" (*National Institutes of Health, 1994*, chap. 3, p. 47). As noted earlier, this choice of the PS as a research limit was non-unanimous, and it is also true that concern over the possibility of pain or sentience was not the exclusive reason for choosing the PS as a research limit, as arguments suggesting that pre-PS embryos could not yet be human individuals due to their capacity to twin and aggregate were also influential (*National Institutes of Health, 1994*, chap. 3, p. 47). But putting aside these points for now (and we will take them up again below), the operative reasoning in the choice of PS is clear: The prospect that embryos might experience pain or sentience during an experiment is morally concerning, and the PS was identified as reliably preceding the development of this capacity during canonical embryogenesis: Thus, setting a research limit at the appearance of the PS prevents researchers from putting an embryo in this morally concerning situation.

The basic problem SHEEFs pose is that, by dint of their ability to recapitulate embryonic development, they could raise moral concerns comparable to non-synthetic embryos, but a research limit based on canonical development that works in embryos to avoid the concerning situations might be ineffective for SHEEFs because they need not develop canonically. For instance, through the methods described above, researchers could soon find ways to generate SHEEFs that proceed through neurulation without having first gone through a PS by differentiating hPSC into cells of the three germ layers and patterning them with appropriate signaling centers *via* 3D printing into a synthetic PS-less analogue of an embryonic disk. Like neurulating non-synthetic embryos, these PS-less neurulating SHEEFs would be created in a condition that could be morally concerning because they could have the possibility of experiencing sentience or pain. But while the 14-day rule would prevent this occurrence for the non-synthetic embryo, it would be completely ineffective for the SHEEF because the rule's triggering condition – the appearance of a PS – would never occur. Moreover, adjustments to the 14-day rule that might alter the research limit for embryo experiments to a later (or earlier) stage would not serve to correct this problem, for while the revised limit might specify different features than the PS as points at which experiments should be terminated, synthetic biology-based tissue and cell engineering methods could still make it possible to bypass these features, just as they might enable bypassing the PS. In short, the general problem raised by SHEEFs is that, given the many emerging technical options for generating them and their expected developmental plasticity, research limits that are triggered by entry into any particular stage of canonical embryogenesis may lose their effectiveness. The situation can be described figuratively by saying that current guidelines look at development as a single long highway, so that boundaries that address moral concerns can be set by erecting a stop sign at a suitable place. But synthetic biology is now making it possible to construct SHEEFs that may travel 'off-road' or find previously unmapped alternative paths that can carry them around the stop sign into the territories to which the sign was supposed to restrict access.

It may yet be questioned whether the developmental plasticity of SHEEFs will prove to be so great as to permit neurulation to take place without ever going through a PS stage. It is therefore worth noting that multiple pathways may lead to comparable endpoints. First, putting aside SHEEFs that proceed through germ layer formation such as the gastruloids of *Warmflash et al. (2014)*, methods of generating cerebral organoids are improving rapidly and are leading to an increasing ability to model diverse brain regions (*Jo et al., 2016*; *Mason and Price, 2016*), and generation of nociceptors from hPSC has also been demonstrated (*Chambers et al., 2012*). With continued progress, additional neural elements may be generated and combined into human entities that possess operative and sustainedly active human neural pain sensation pathways. Second,

we cite our lab's experience with a completely different approach that suggested that human entities could be created without passage through a PS that would raise moral concerns were they to arise in embryos. In 2011 we conducted experiments on what we called Embryo Scaffold-Aided Tissue Engineering (ESATE), which involved injecting human induced pluripotent stem cells (hiPSC) into previously decellularized and fixed mouse embryos (*embryo scaffolds*). Our hypothesis was that signaling factors remaining in the decellularized matrix would induce the hiPSC to develop into multiple human cell types at once throughout a scaffold. We originally conceived ESATE as a way of developing many cell types and tissues at once from hiPSC, but soon came to realize that these experiments, if successful, could result in embryo-like entities if sufficient tissues developed and became functionally interconnected, e.g., if brain and heart primordia emerged and the heart began to deliver nourishment to the primordial brain. Perceiving that the creation of such entities might be morally concerning, we scrutinized established guidelines for direction and then asked for counsel from our institution's Embryonic Stem Cell Research Oversight (ESCRO) committee. After careful study, the committee ultimately found that the experiments violated no existing guidelines but asked us to keep them abreast of further developments. Subsequently, we ended these experiments in large part due to our inability to adequately perfuse the embryo scaffolds, a problem that now promises to be addressable in organoids and SHEEFs through vascularization. Unlike SHEEFs, the ESATE experiments raised issues other than the 14-day rule for current ethics guidelines—notably, whether injection of hPSC into decellularized animal embryo scaffolds might be subject to rules for chimeras—but both cases highlighted the same problem with existing guidelines: that research limits that work to avoid morally concerning situations for embryos (or chimeras) might fail to be effective for human entities created by new technologies that could re-organize development and avoid the biology specifically designated in the limit.

## 3 Conceptual and structural difficulties that impede extension of current guidelines to SHEEFs

Above we found that a key problem with the ability of the current or adjusted versions of the 14-day rule to apply to SHEEFs is the rule's specification of a stage from canonical embryogenesis as a research limit. Indeed, canonical embryogenesis acts as a central conceptual model for the entirety of current embryo and stem cell guidelines, and this is a natural outgrowth of a history that started with a focus on the research and development of assisted reproductive technologies by the 1979 US Department of Health, Education, and Welfare's Ethics Advisory Board (*Ethics Advisory Board, 1979*). The scope of research covered by the guidelines expanded with succeeding commissions as techniques and biological entities considered by their predecessors to be futuristic or in early development became reduced to practice and usable as tools of research. Thus, Somatic Cell Nuclear Transfer (SCNT), which had not yet been demonstrated as of the 1979 Ethics Advisory Board or the 1984 Warnock committee, was mentioned by both as a method under development (*Walters, 1979*, pp. 44ff, 51–52; *Warnock, 1984*, section 12.14, p.73), but was not the subject of any specific recommendations. But in 2005, after SCNT had been demonstrated in animals and hESC had been derived, the US National Academy of Sciences (NAS) formulated guidelines (*National Research Council, 2005*) for both. Research limits involving parthenogenesis and androgenesis had a similar history and were ultimately integrated with the NAS SCNT and hESC limits in the same 2005 guidelines (*Warnock, 1984*, Section 12.10, p. 72; *National Institutes of Health, 1994*, p. xv; *National Research Council, 2005*, Section 4.5). The NAS added limits for hiPSC in 2008 (*National Research Council, 2008*). However, throughout this gradual process of expansion, the guidelines retained their original perspective on development as the progression starting from zygotes through the canonical series of embryonic stages, i.e., through first divisions, morulae, blastocysts (all achievable in culture), and onward through later stages after implantation. Unlike zygotes, the newer cell types might not be able to progress through the complete series and might need conditions beyond what zygotes required to activate and sustain their development: For instance, parthenotes generally arrest after early divisions but can sometimes achieve the blastocyst stage (*McElroy et al., 2008*; *Versieren et al., 2010*), but hPSC have potential to progress through the complete series after injection into blastocysts (an experiment prohibited by guidelines that has never actually been reported as having been conducted). For brevity, we will call all these cell

types (including zygotes) *embryogenerative* cells and the ranges of canonical embryonic stages they can achieve as their *sequelae*.

A consequence of this continued reliance on canonical embryogenesis in current guidelines is that almost all research limits are operationalized in terms of procedures with embryogenerative cells and their sequelae, not just the 14-day rule. For instance, just as the 14-day rule is specified by referring to an action (termination of an experiment) that should take place at a canonical embryonic stage (appearance of the PS), the recommendations in the NAS guidelines from 2005 to 2010 concerning the generation of embryos using hESC are operationalized in terms of specific procedures with embryos and embryogenerative cells, e.g., that hESC should not be injected into human blastocysts or into non-human primates at embryonic or other stages (*National Research Council, 2005*, rule 1.2(c); *National Research Council, 2008*, rule 7.3(c); *National Research Council, 2010*, rule 7.3). The NAS research limits involving SCNT, parthenogenesis, and androgenesis are similarly operationalized as a recommended prohibition against the implantation of blastocysts made using these embryogenerative cell types and sequelae (*National Research Council, 2005*; *National Research Council, 2008*; *National Research Council, 2010*, rule 4.5). Starting in 2005, NAS began referring to broader classes of entities and criteria in some research limits—for instance, on experiments that might lead to significant integration of human germ cells or neurons into corresponding tissues of human-animal chimeras in *non-embryonic* stages of development—but these limits were still to be applied to procedures with embryogenerative cells and sequelae at earlier embryonic stages, at the point when hPSC-derived human germ cells or neurons might be injected into animal embryos (*National Research Council, 2005*; *National Research Council, 2008*). Similarly, in 2010 the NAS for the first time specified a research limit on usage of a *non*-embryogenerative cell type: the multi-potent human neural stem cell (*National Research Council, 2008*, Section 7.4). But just as with the 14-day rule, any research limits that delimit operations with embryogenerative cells and canonical sequelae have the potential to become ineffective for SHEEFs, if these SHEEFs are generated by methods that do not involve these cells or sequelae.

This operationalization of research limits based on canonical embryogenesis has a structural correlate that additionally complicates the extension of current guidelines to SHEEFs. Since the number of embryogenerative cell types is small and the set of canonical sequelae is likewise limited, it is easy to organize research limits into lists of operations that are considered permissible, impermissible, or that should receive further review. But compared to the limited universe of operations with these cell types and sequelae, the space of the cell and tissue engineering operations that could be used to generate SHEEFs is vast, combinatorially complex, and rapidly growing, and may not be amenable to simple enumeration. Thus, current guidelines can specify essentially all of their research limits in terms of entities such as *hPSC*, *blastocyst*, and *intact embryo*, and a small list of actions such as *culturing*, *injection into a blastocyst*, and *implantation into a uterus*. But if we wanted to similarly operationalize a research limit that would prevent the creation of a PS-bypassing SHEEF that could experience pain, we would need a much more complex ontology that might include such entities as *hPSC-based inducible neuron* (*Busskamp et al., 2014*) or *layered array of hPSC-derived endoderm, mesoderm, and ectoderm progenitors*, and might refer to actions such as *patterning through 3D printing* or *cell deposition on microfabricated surfaces*. Indeed, even without considering SHEEFs and the engineering methods that might generate them, the specification of research limits in terms of operations on embryogenerative cells and their sequelae has already become problematic. One sign of this is that since 2005, NAS guidelines (*National Research Council, 2005*; *National Research Council, 2008*; *National Research Council, 2010*) have contained two separate statements of the 14-day rule, one in connection with embryo culturing (rule 1.2(c) in 2005; 1.3(c) in 2008 and 2010), and the other in connection with cells and embryos created through SCNT, parthenogenesis, and androgenesis (rule 4.5). Thus, even extending the guidelines to cover a small set of new embryogenerative cell types led to complication of their structure. (Notably, in the ISSCR's 2016 guidelines, research limits have been organized so as to contain a single statement of the 14-day rule (*International Society for Stem Cell Research, 2016*; Recommendation 2.1.3.3), and the specification of the different varieties of embryos and embryo-like structures covered under it is presented in a complex Glossary. This organization eliminates the duplication of the

rule but is still challenged by SHEEFs that bypass stages like the PS.)

## 4 Basing research limits for SHEEFs directly on the moral status signifying features: a call for exploratory inquiries

Given the nature of the problems raised by SHEEFs for current guidelines, we see only one solution: Instead of tying research limits to stages of canonical embryogenesis in an attempt to preempt SHEEFs from being generated in morally concerning conditions, limits should be based as directly as possible on the appearance of features or capacities that are associated with emergence of moral status. This strategy immediately avoids the problem of SHEEFs that might bypass preemptive limits defined by canonical stages like the PS. On our proposal, the research limit would be set at the first entry into the condition that directly raises moral concern—in this case, the appearance of neural substrates and functionality required for the experience of pain—rather than at a canonical preemptive stage (the PS) that, as noted earlier, is not seen as intrinsically morally significant. Such a reframed research limit would be immune from the problem of bypass because it is set at the condition that *directly* raises moral concern and it would function correctly no matter whether this condition was approached *via* canonical or non-canonical development. Moreover, as possession of these neural substrates would be sufficient to trigger application of the limit regardless of what technologies were used to generate them, this strategy avoids the need to specify the limit through a list of the many human cell type and tissue engineering procedures that might be used to generate them. In our figurative portrayal above, we described current guidelines as attempting to prevent entry into morally concerning areas by erecting a stop sign on what it views as a single long highway of embryo development, a stop sign that could be evaded by synthetic biology methods that are finding alternate paths of development and enabling 'off-road' travel. Extending this portrayal, our proposal views development not as a highway but as a landscape in which particular territories are defined by the possession of moral status by developing embryos or embryo-like entities, and it aims to protect these territories by erecting perimeter fences around them, a scheme that is illustrated in *Figure 1*.

The critical question that now arises is how these territories associated with moral status at a level sufficient to require a research limit can be identified. While the complex landscape defined by SHEEFs and engineering makes this task especially novel and challenging, the commissions that developed the 14-day rule provide an example for how such deliberations were conducted for non-synthetic embryos, and their example could be followed with suitable modifications for SHEEFs. Specifically, these commissions started with a broad discussion about biological features and capacities whose emergence in canonical non-synthetic embryo development they deemed morally significant. But because these features and capacities arise in a canonical sequence, they could prevent experiments involving *any* of them through a single research limit at the *first appearing* one: Thus, once collective agreement was obtained on this feature, broad discussion was curtailed and focus was narrowed to specifying a preemptive research limit for that feature. To adapt this process for SHEEFs, the initial broad discussion phase should instead be expanded to a broad effort to assemble a catalog of morally significant embryonic features and capacities without abridgment or preferential attention to the first appearing feature, and follow-on analysis should aim at defining the biological substrates that underpin each of these features individually rather than selecting preemptive canonical states preceding them. However, for SHEEFs an important additional phase lies between these two activities: It must be assessed which of the features (individually and in combination) that are morally significant in embryos are also morally significant in SHEEFs. We propose that an effort to undertake this process might best be organized as an extended set of exploratory scientific, philosophical, and sociological inquiries led by the research and bioethics communities *vs.* a commission assigned to determine guidelines and research limits. In sections 4.1-4 below, we offer high level suggestions on each of the three phases and on the structure of our proposed exploratory effort. But before proceeding, we note that because this process starts with an exploration of how embryos and SHEEFs may cross thresholds of moral significance as they develop biologically, it must pay close attention to viewpoints that take moral status to be developmentally emergent. We will introduce terminology below to clarify what information we believe must be extracted from these views. We stress that this focus does not mean that

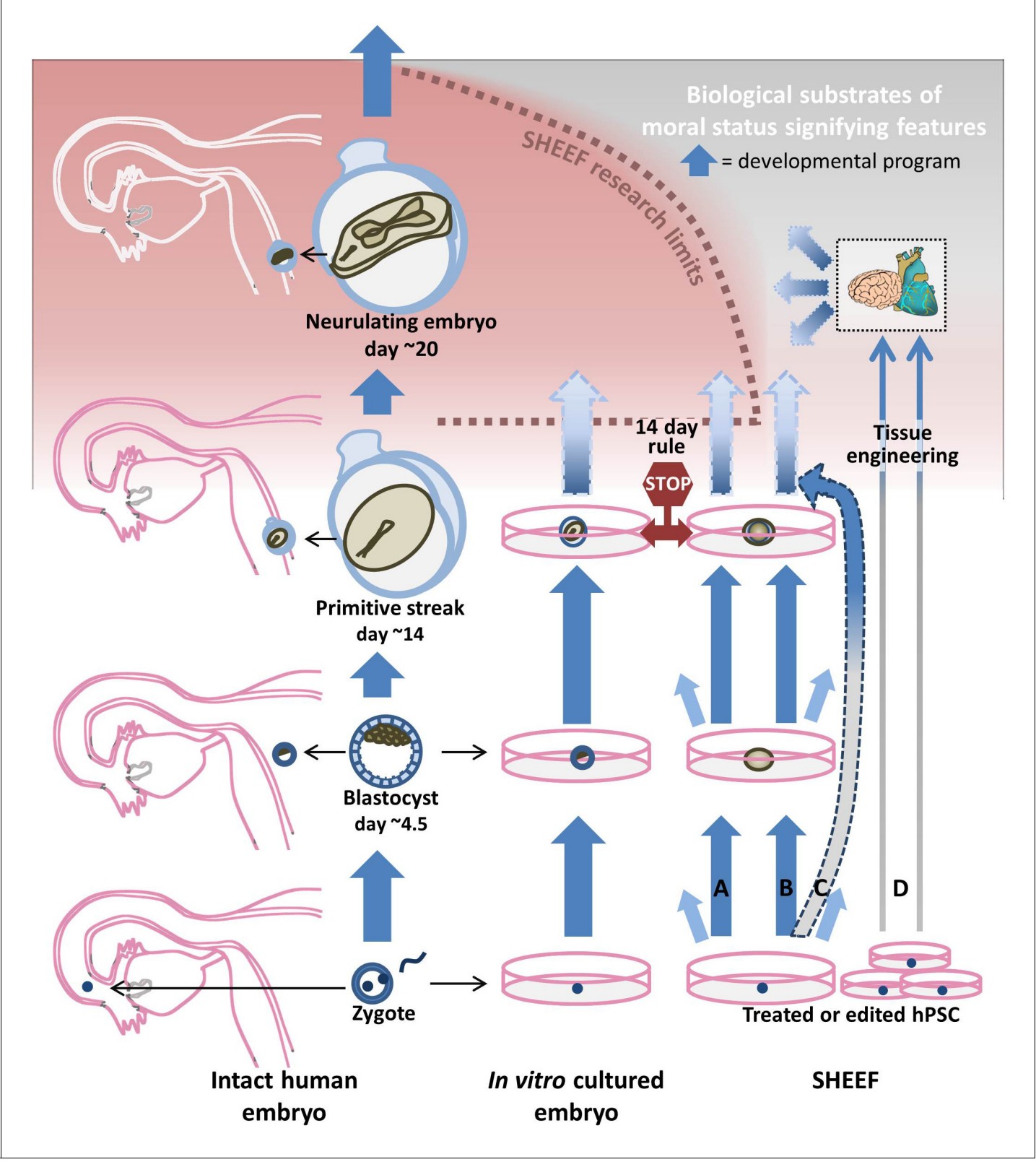

**Figure 1.** Biological landscape of embryos and *Synthetic human entities with embryonic features* (SHEEFs) in relation to moral status. Embryos derived through sexual intercourse or assisted insemination (*Cantineau et al., 2013*; *Hurd et al., 1993*) (left), cultured embryos (center), and SHEEFs (right) start from types of pluripotent cells (zygotes and hPSC; bottom) that have different capacities for development: Embryos formed from zygotes derived sexually can develop into fetuses in utero (vertical arrows, left). Embryos can also be generated from zygotes formed in vitro and these can also result in

*Figure 1 continued on next page*

*Figure 1 continued*

normal fetuses upon implantation (vertical arrows, center); however, the course of further development in culture is uncertain if implantation does not take place (fading blue arrow, center top), and such experiments are forbidden for ethical reasons (*Deglincerti et al., 2016*; *Shahbazi et al., 2016*; *Weimar et al., 2013*; 14-day rule). Though pluripotent, hPSC exhibit very different courses of development from zygotes that depend on the treatments and culturing conditions and are subject to stochastic effects (short light blue, multi-directioned arrows, right), presumably because they start in a different molecular state from zygotes and do not accurately receive and deliver normal developmental signals. Methods that impose external patterning on hPSC colonies appear able to partly compensate for these deficits and result in SHEEFs that recapitulate more aspects of canonical embryonic development (dark blue A and B vertical arrows, right), as seen in recent experiments by *Warmflash et al. (2014)* that yielded a version of a Primitive Streak (PS), a developmental feature of early embryogenesis. More sophisticated hPSC cell and tissue engineering methods are expected both to generate SHEEFs that model development more accurately, but also those that develop non-canonically into entities very different from embryos. Meanwhile, a long series of commissions (*National Institutes of Health, 1994*; *Warnock, 1984*; *National Research Council, 2010*) has considered the ethics of experimental research on cultured embryos. These commissions have generally (but non-unanimously) seen the moral status of embryos as increasing (red gradient, top) as they develop the biological substrates of features and capacities taken to be morally significant (moral status signifying features; see text). The 14-day rule arose from basing the research limit for embryo experiments on the first such features to arise in canonical embryological development—the emergence of brain rudiments during neurulation at ~14 days that could, with further development, allow embryos to experience pain. To build in a safety margin the limit was set at the PS, which immediately precedes neurulation in canonical embryogenesis. The fact that the entities produced by *Warmflash et al. (2014)* have a PS has now led to questions about whether SHEEFs might need to be regulated under the 14-day rule (see text). But while some SHEEFs could model canonical embryo development so accurately that they develop a PS and subsequently neurulate (right, vertical arrow A, which leads into the red zone), this may not be so for others (right, vertical arrow B, which leads into a gray zone with no determined moral status), and it is also possible that sophisticated engineering could produce SHEEFs that bypass the PS entirely but still develop neural substrates (right, vertical arrows C and D). Thus, because a PS may neither be necessary nor predictive of whether a SHEEF could develop a brain rudiment that eventually could experience pain, the 14-day rule may be ineffective for SHEEFs. To overcome these difficulties, we propose that research limits for SHEEFs be based as directly as possible on the biological substrates of moral status signifying features, instead of on canonical embryonic features like PS that do not have intrinsic moral significance. Whereas current guidelines based on canonical embryogenesis assume that a research limit at an early canonical embryonic stage will prevent development of any ensuing morally concerning feature by setting a 'stop sign' on a single 'highway of development' (red stop sign), our proposed strategy treats morally concerning features as 'territories in development' and recommends that research limits be set like 'perimeter fences' that prevent entry into them from any direction (red dotted boundary line). Note that the red dotted line in this figure is only meant to depict how a protecting perimeter might be set up, and does not represent our views of where it ought to be set up. The depiction of the female reproductive organs used in this figure was taken from the figure "Human Fertilization" by Ttrue12 under license CC-BY-SA-3.0.

countervailing views must be excluded from consideration, a point we will return to in section 4.4.

### 4.1 Cataloging morally significant events in canonical embryogenesis

The commissions that developed and endorsed the 14-day rule all commented that many different views existed regarding how an embryo's moral status might change over the course of its continuously developing biology. But because they only needed to collectively agree on the earliest appearing feature associated with such a change to meet their goal of setting limits on embryo experiments, their consideration of later features was brief. Indeed, beyond its identification of the capacity to experience pain as the key moral concern for determining the length of embryo experiments, the Warnock report's only mention of other views were the above cited comments about the "wide range of opinion" on this issue (*Warnock, 1984*, 11.20 p. 65; see also Section 11.15, p. 62). But the 1994 RHERP committee reported more of their discussion of other features. Thus, their report cites sources that recognized alternative (and later) benchmarks of neural development than sentience, such as "the beginning of brain activity or brain function" and "well-developed cognitive abilities such as consciousness, reasoning, or the possession of self-concept" (*National Institutes of Health, 1994*, chap. 3, p. 37). It also cites "often mentioned" features of very different natures such as "human form, capacity for survival outside the mother's womb, and degree of relational presence (. . . to the mother herself or to others)", and documents that at least one RHERP panelist named "the onset of heartbeat at day 22" as a specific event that signified increased relational presence (*National Institutes of Health, 1994*, chap. 3, p. 38 and p. 47). On a completely different level, the RHERP committee also acknowledged two abstract capacities with a long history of discussion in the embryo ethics literature—the entry into a state of individuality associated with the loss of the ability of embryos to twin or aggregate (*Diamond, 1975*; *Donceel, 1970*; *Gilbert, 2008*; *Montague, 2011*; *Munthe, 2001*), and the developmental potential of zygotes and

early embryos (see, e.g., *Stier and Schoene-Seifert, 2013*)—but did not appear to see these as decisively supporting the PS as a research limit (*National Institutes of Health, 1994*, chap. 3, pp. 36-40). Like the earlier Warnock committee they focused on sentience and endorsed the former's choice of the PS as a limit on the basis of the need for clear policy.

To extend these deliberations to SHEEFs, the discussions by these and other prior commissions could be revisited and then expanded to both a deeper probing of the views they mentioned and a broad sampling of viewpoints they may not have considered. The goal should be to come up with a catalog of features, capacities, or events in embryo development that have been articulated by established or other worked-out viewpoints as points at which embryos require increased moral respect, attention, or protection. Here we broadly call these all *moral status signifying features* (or simply *status signifying* or *signifying* features where the context is clear). We stress that our call for identification of such features does not imply that we think moral status can be conceptually reduced to them: The relation is one of articulated association, not existential identity.

### 4.2 Determining the applicability of the catalog to SHEEFs

Any features collected in the catalog in section 4.1 would be considered moral status signifying for non-synthetic embryos by at least some viewpoints, but possibly not for SHEEFs. The problem of determining which (if any) features might qualify for such status, and to what extent, will require inquiries of a sociological and philosophical character. On the sociological side, representatives of viewpoints included in the catalog could be probed directly for moral concerns or reactions to the occurrence of these features in possible SHEEFs. Importantly, it would be necessary to probe for reactions to combinations of signifying features, including combinations that do not naturally co-develop in canonical embryogenesis. Obtaining well-defined reactions may be difficult because the prospect of such non-canonical SHEEFs may be so novel that established viewpoints may be unprepared to offer articulated responses to them. Nevertheless, a spectrum of reactions is predictable. For instance, it is likely that SHEEFs that are engineered to be very like non-synthetic embryos in features and potential ("embryos in a dish" as per *Pera et al., 2015*) will excite moral reactions comparable to those raised by

embryos, while novel combinations of features, such as a SHEEF that combines a human beating heart and a brain that lacks the capacity for pain and sensation, will raise more uncertain reactions. It can also be expected that some combinations of signifying features could evoke "yuck" responses (*Brown, 2006*; *Niemelä, 2011*) that fall short of articulated moral objections: We speculate that SHEEFs with recognizable human form but that lack other features might evoke such reactions, such as a human-appearing SHEEF with a beating heart but no brain. In any case, every reaction – positive, negative, or uncertain – will provide a "data point" that will be pertinent to the question of what research limits might be set for SHEEFs, and the very process of probing for reactions could stimulate discussions internal to these viewpoints that could lead to more consolidated opinions about their moral status. A historical analogue might be found in the broad span of initial reactions to, and subsequent positions taken on, hESC by social, cultural, and religious traditions around the world, as they integrated these novel entities into their pre-existing moral frameworks (*Walters, 2004*).

On the philosophical side, two kinds of inquiry could be considered. First, efforts could be made to explore the extent to which "data points" derived from the probing of viewpoints above could be aligned into systematic conceptual frameworks concerning the moral status of SHEEFs. Second, moral status signifying features of embryos that have been elaborated in the philosophical literature, such as developmental potential and entry into a state of individuality, could be reviewed with an eye to whether and how they might apply to SHEEFs, and possibly extended accordingly. Indeed, a conceptual framework may need to be devised to define the meaning of developmental potential in SHEEFs and ways of operationally measuring it, as SHEEFs will have only abridged abilities to achieve the canonical developmental endpoint of non-synthetic embryos, i.e., into independent, free-living organisms (unless they are specifically engineered to do so) (*Guenin, 2008*; see chap. 2, pp. 29-30 and chap 8.4). Also, as the tissue printing and patterning methods used to generate SHEEFs could potentially be used to aggregate them or clone them by splitting, the arguments linking the loss of capacity for twinning and aggregation with moral status and the individuation of embryos may have to be re-evaluated for SHEEFs.

### 4.3 Identifying the biological substrates of moral status signifying features

Any combinations of features found to be moral status signifying for SHEEFs would be candidates for corresponding research limits. But to follow our proposal to base these limits as directly as possible on these features will require that the biological substrates that underlie their functionality be specified to the extent possible. The ability to do this will depend on the depth of biological knowledge available regarding each signifying feature, and on the degree of biological concreteness of the feature itself. For concrete biological features, it may be possible to specify in detail what kinds and organizations of tissues need to be present and functional: For instance, if a beating heart were identified as moral status signifying for SHEEFs, a fairly detailed map of the embryonic cell types and structures that participate in cardiogenesis has been developed in mouse (*Brade et al., 2013*) that could be integrated with embryological, cell culture, and molecular data from human. For abstract features such as developmental potential, the corresponding biological substrates might only be identifiable after undertaking the conceptual analyses into the meaning of this potential for SHEEFs described in section 4.2. Other features would likely fall between these extremes: For instance, neural pathways that underlie pain perception have been mapped out at high levels, such as the pathway from nociceptors through the spinothalamic and thalamocortical tracts into the somatosensory cortex (*Al-Chaer, 2012*), but uncertainties remain about the composition and functionality of these pathways in embryos (*Lee et al., 2005*; *Lowery et al., 2007*). Caution would also be needed regarding the possibility of generating SHEEFs with a central pain syndrome in which pain could be experienced without the complete normal pathway. Finally, the definition of the biological substrates associated with some status signifying features, like the acquisition of recognizable human form, might need to be approached more by sociological *vs.* biological methods, as the perception that a SHEEF has human form could differ across viewpoints or cultures. In basing possible research limits on the biological substrates of moral status signifying factors, the most useful information would be the identification of substrates and functionalities that are jointly necessary for the feature's presence and operation. Research limits could then be framed by specifying subsets or threshold levels for these features and functions that must not be allowed to appear jointly in a SHEEF. Appropriately configured subsets and thresholds would allow safety margins to be built into the limits against the possibility of generating SHEEFs in morally concerning conditions. For example, a research limit for blocking creation of SHEEFs that have the ability to experience pain might require that at least *two* forms of neurons in the pathway from nociception to cortex be absent or non-functional, which would provide a safety margin against the unanticipated generation of one of these forms because pain could still not arise without the other. (We stress that we have presented this example for purposes of illustration only and not as a specific proposal.)

### 4.4 Organizing the inquiries

It is because of the novelty of SHEEFs and the many open questions regarding how to assess their moral status that we propose a set of exploratory inquiries into their moral and scientific issues rather than a commission for determining guidelines and research limits for them. Guidelines and research limits may ultimately be desirable and needed, but a commission will work best only when enough such groundwork has been done to provide it with systematic information, analyses, and alternative positions. Once this in hand, a commission could be assembled along the lines of prior embryo and stem cell commissions with the main goal of coming to a collective agreement on guidelines and research limits, as well as any needed *non-*research limit guidelines (such as requirements for informed consent). The inquiries suggested here differ from a commission by the *absence* of a collective agreement goal: If the explorations stimulated divergent formulations for handling the issues, this outcome would be welcome and provide important alternative positions that the commission could consider. If these inquiries are undertaken, it will be important to monitor the emergence of coherent formulations as an indicator of when it may be time to start organizing a commission.

We have suggested that the research and bioethics communities lead these inquiries because they have the most immediate contact with the scientific and ethical issues being raised by SHEEFs. We will have more to say about what bodies within these communities might assume this role in section 6 below. Here we confine ourselves to three suggestions on how these might be structured: (1) While the research

and bioethics communities might lead this effort, it should be multinational and explicitly reach out to other disciplines, communities, institutions, and traditions in order to get broad input and stimulate discussion on the appropriate handling of these issues in ways exemplified in (but not restricted to) sections 4.1, 4.2 and 4.3. While as earlier noted, close attention must be given to viewpoints that accept moral status as developmentally emergent, it is highly important that input and involvement also be solicited from opposed viewpoints. When the time comes for a commission, it will be vital for it to include these views along with those that endorse developmental emergence and to negotiate between them in the process of coming to the collective agreement, in the same manner as has been done by all prior commissions, for its recommendations to be respected and adopted. (2) The exploratory character of these inquiries and the likely need to prompt participants to move from uncertain initial reactions to more formulated viewpoints suggests a need for creative mechanisms for engaging participants. In addition to such standard venues as forums and conferences, means could include themed journal issues and issuing open challenges on particular questions. For instance, scientists could be challenged to generate a synthetic embryonic disk *in mouse* using tissue and stem cell engineering that could neurulate without passing through a PS, and ethicists could be challenged to consider whether (and propose conditions under which) generating a pain-sensing human SHEEF might be ethically permissible, given the high scientific and medical importance of understanding human pain. (3) A starting point for the organization of such inquiries is to assemble a list of issues and questions that could be addressed by these means. Our suggestions here in section 4 and comments in section 5 below might be used as an initial draft of such a list.

## 5 Interfaces with other ethical issues and ethics processes

In section 4 we laid out proposals for addressing what we consider to be the core ethical issues raised by SHEEFs, but these issues interface with other ethical issue areas and processes. Here we describe three such interface areas that might be studied along with the core issues within the exploratory inquiry process we have proposed, with an eye to how ethical recommendations might be harmonized across their boundaries.

### Reassessment of the 14-day rule

As noted earlier, the *Warmflash et al. (2014)* experiments demonstrating a SHEEF with a PS have prompted interest in the possibility of revising the 14-day rule to allow more extended experiments with non-synthetic embryos. Our proposal that research limits for SHEEFs should be based as directly as possible on moral status signifying features might effectively revise this rule for SHEEFs by moving the research boundary to (for instance) the development of functional neural pain circuitry. The question arises whether this proposal could or should be applied to non-synthetic embryos. We suggest that the answer is: Not necessarily. Our proposal to base research limits for SHEEFs directly on signifying features is based on the inference that, given the engineering methods used to create SHEEFs and their potential for developmental plasticity, revising limits in this way will be the only workable way to prevent the creation of SHEEFs in morally concerning conditions. But non-synthetic embryos go through the PS stage routinely and are not generally developmentally plastic in this way, so this conclusion does not follow. A more secure conclusion would be that, if *for independent reasons* the revision of the 14-day rule for embryos is justified, the considerations we have outlined for SHEEFs might be relevant to what new limit might replace it. But this does not imply that the neural substrates underlying pain sensation would be the right 14-day rule replacement for embryos because they might develop a different status signifying feature before the ~20 weeks (gestational age) needed for development of neural substrates of pain (*Lee et al., 2005*), and also because the 14-day rule has also been defended for embryos by reference to capacities for twinning and developmental potential whose status is uncertain for SHEEFs (although these arguments have been challenged for embryos, too: see comments and citations above). Finally, it is possible that many signifying features in the catalog of status signifying features for embryos developed *via* 4.1 could be deemed non-signifying for SHEEFs by the process in section 4.2. Features whose moral significance for embryos involves *relational presence* would seem especially amenable to being deemed non-signifying for SHEEFs (see section 4.1 above).

### Impacts of using SHEEFs to understand human embryogenesis

To realize the promise of SHEEFs as systems for analyzing human embryogenesis experimentally will require generating SHEEFs that are as close as they can be to non-synthetic embryos without triggering the restrictions that would apply to them. This will increase pressure for guidelines and research limits for both SHEEFs and embryos that give precise definition to how SHEEFs must be configured to avoid making them morally equivalent to embryos, aggravating the problems noted by Pera et al. (*Pera et al., 2015*) regarding how the different definitions of "embryo" in place in different jurisdictions today will complicate research using SHEEFs. It will also increase the demand for experiments with non-synthetic embryos that go up to the permissible boundaries of embryo research, as knowledge gained from these would enable production of more developmentally accurate SHEEFs. Regarding the technical problem of how to create SHEEFs that come very close to but avoid crossing ethical boundaries, SHEEFs could be generated using hPSC that are specifically engineered in ways that will prevent the SHEEFs from developing a cell type or function essential to a moral status signifying feature, but which would follow embryonic development as closely as possible in every other respect. Historically, this is analogous to the "altered nuclear transfer" (ANT) method (*Hurlbut, 2005*) put forth during the formulation of United States' policies on the derivation of hESC (*President's Council on Bioethics, 2005*). ANT involved using SCNT with nuclei engineered with mutations that would generate "compromised life forms" (*Guenin, 2005*) that could not develop to the PS stage. ANT was never accepted into the United States' hESC guidelines, and the argument that it was ethically better than deriving hESC directly from embryos was debated (*Guenin, 2005*). The ethics of using an analogous strategy to generate SHEEFs for the study of human embryogenesis might be less problematic because SHEEFs start from guideline-approved hPSC and are generated using methods that already depend on cell and tissue engineering that involve neither creation nor destruction of intact embryos. Nevertheless, it would be good to re-examine the ethical arguments over ANT for their relevance to SHEEFs.

### Ethical and conceptual boundary issues

As synthetic *human* entities, SHEEFs are exclusively human by definition and we have focused on research limits that involve entirely human tissues and development, such as the 14-day rule. But as the techniques used to generate SHEEFs can also be applied to non-human cells and tissues, this focus is to some extent arbitrary. There is no technical reason why cells and tissues developed from PSC from non-human animals, or mixtures of PSC-derived human and non-human cells and tissues, could not be used to engineer entities that undergo aspects of embryonic development. Entities created from mixtures of human and animal cells and tissues would raise moral concerns similar to those raised by chimeras, while those created exclusively from animal cells could raise issues related to animal welfare: For instance, a synthetic animal entity that presented operative and sustainedly active neural pathways for pain might excite moral concerns similar to those covered by guidelines that require minimization of pain and distress in experiments with animals (*National Research Council, 2011*). Similarly, as SHEEFs are by definition synthetic entities with *embryonic features*, our focus has been restricted to ethical issues that arise in embryonic contexts. But we have already noted that synthetic biology methods might be used to generate *post-embryonic* human cerebral and neural tissue organoids that likewise present complete and active pain pathways, possibly even in childhood or adult forms. Beyond sentience and pain, ethical concerns have been raised in both biological (*Greely et al., 2007*) and non-biological (*Ashrafian, 2016*; *Coeckelbergh, 2010*) contexts with the possibility of generating entities with consciousness and self-awareness, and these concerns could potentially also be triggered by very sophisticated brain organoids. Moral concerns could also arise with other types of human entities, as illustrated by our ESATE experience above.

## 6 Discussion

We and others (*Daley et al., 2016*; *Denker, 2014*; *Hyun et al., 2016*; *International Society for Stem Cell Research, 2016*; *Pera et al., 2015*; *Warmflash et al., 2014*) have noted that rapid advances in human stem cell and tissue engineering are leading to the ability to generate synthetic human entities that exhibit embryonic features (here called SHEEFs) that are raising challenges to currently established ethical guidelines for embryo and

stem cell research. While the "gastruloids" generated by the Warmflash et al. experiments—the most advanced SHEEFs to date—have come closest to recapitulating an embryonic feature specifically covered in current guidelines, these entities are still very remote from actual embryos. But given the many available methods now maturing and new ones under development, there is little doubt that it will soon be possible to generate SHEEFs that exhibit more and later embryonic features, and where these features more accurately represent those of embryos. As researchers who have been active in the development of these and other (*Baltimore et al., 2015*; *Esvelt et al., 2014*; *Oye et al., 2014*) new technologies, we feel we have a deep responsibility to call attention to the potential ethical, social, legal, or environmental consequences of such technologies early—before these consequences are actually experienced—and seek broader institutional, community, and societal guidance. We have already sought institutional guidance from our ESCRO committee on two occasions, once for our ESATE experiments of 2011, and more recently for microfabricated hiPSC experiments similar to those of Warmflash et al. (which we have not described here). In both cases our ESCRO committee recognized the problems and was keenly interested, but could only offer limited guidance because the policies on which they themselves relied did not cover the kinds of experiments we were conducting. Our goal in writing this article has been to call these issues to the attention of a wider audience and begin to engage its collective energy and creativity towards filling this gap in guidance. We have thus called for the research and bioethics communities to lead a set of exploratory inquiries to gather input on moral concerns triggered by SHEEFs, and to consolidate thinking about how these might be addressed. Ultimately we should like to see a commission established to set guidelines for SHEEF research, but we have argued that this should be done only after the exploratory inquiries have laid this informational and conceptual groundwork. We feel strongly that this is the proper time to start this work, even though SHEEFs are still very distant from embryos, for the absence of a scientifically successful experiment is no barrier to productive examination of their ethics. A precedent for this is provided by the human neuron mouse (*Greely et al., 2007*), which resulted in changes to the NAS stem cell guidelines (*National Research Council, 2008*;

*National Research Council, 2010*, rules 7.3-4) even though the experiments that led to these ethics processes had never actually been conducted (*Greely et al., 2007*; *Weissman, 2005*).

While the ethical issues raised by SHEEFs have called attention to the 14-day rule (*Denker, 2014*; *Hyun et al., 2016*; *Pera et al., 2015*), we have argued here that the fundamental problems lie more deeply in the conceptual and structural assumptions of current guidelines, and that changing the 14-day rule would not address the these problems. The core of our proposed exploratory inquiries lies in our analysis of what these fundamental problems are and what can be done to address them. In particular, we have argued that addressing the issues raised by SHEEFs will require research limits that are based as directly as possible on the presence of early forms of embryonic features that signify moral status. In contrast, embryo and stem cell research limits are currently based either on the attainment of particular canonical developmental stages in intact embryos (such as the PS), or on the use of laboratory operations involving the implantation of natural or modified cells that can generate embryos, or these generated embryos themselves, into other embryos or into animals. These methods of formulating research limits will not work for SHEEFs because, with the growing power of synthetic biology to engineer complex tissues and tissue assemblies, it will soon become possible to generate SHEEFs that can bypass canonical embryonic stages through the use of completely different laboratory operations. However, the project of actually attempting to specify research limits for SHEEFs by basing them on features that signify the moral status in embryos presents many conceptual and practical difficulties. The commissions that originally developed and endorsed the 14-day rule encountered both strongly held views about the moral status of embryos anchored in established traditions, and broad divergence about what features of developing embryos held particular moral significance, and propounded the rule as a difficult compromise. Meanwhile, SHEEFs might present novel combinations of such features that are very unlike those of embryos, and are so new and unfamiliar that traditions may be puzzled and unable to offer articulated opinions about them. Thus, we have recommended a multi-tracked exploratory inquiry process that both solicits opinions on how SHEEFs might be morally concerning from a wide range of disciplines, traditions, and institutions, while at the same time challenging researchers, bioethicists,

and philosophers to develop and refine systematic conceptualizations of SHEEF moral status—a combination of approaches that may both feed systematic reasoning with unfiltered input about SHEEFs, and prompt traditions and institutions to consolidate their thinking about them. Only after these articulated views are in hand will it be useful to convene a commission to deliberate and come to a collective agreement on guidelines and research limits for SHEEFs. We also identify other ethical problem areas that interface with SHEEFs that can be pursued in our proposed inquiries and taken up by an eventual commission, including the impacts of SHEEFs on embryo ethics and experimentation, and the boundaries between SHEEF ethics and guidelines for chimera and animal experiments. Such a commission might also have to consider what other kinds of guidelines beyond research limits might be needed for SHEEFs (such as, for instance, informed consent requirements).

It remains to suggest how our proposed exploratory inquiries might be organized and initiated. We submit that groups that oversee embryo research, such as ESCRO or EMRO committees (*International Society for Stem Cell Research, 2016*), as well as authorities such as the Human Fertilization and Embryology Authority in Great Britain and the Embryo Research Licensing Committee in Australia, are well positioned to take organizing roles. These groups already include bioethicists and researchers from a number of disciplines as well as a variety of 'outside' members representing other institutions and traditions, and may already be reviewing experiments involving the kinds of stem cell and tissue engineering methods that can be used to generate SHEEFs. They will likely also be sensitive to the gaps in current guidelines regarding SHEEF experiments and will have a natural interest in efforts to close these gaps. Indeed, triggered in part by the two cases we have brought to them, our institutional ESCRO recently organized a symposium on the ethics and future of the 14-day rule (*Petrie-Flom Center for Health Law Policy, Bioetchnology, and Bioethics at Harvard Law School, 2016*) that included presentations and working group discussions of some of the proposals in this article. Meanwhile, the International Society for Stem Cell Research (ISSCR) is well positioned as a commission that could eventually issue guidelines for SHEEFs (and, indeed, it already has, as noted above *International Society for Stem Cell Research, 2016*). ISSCR could also participate in organizing the inquiries, which

would ensure international participation, and in monitoring their progress for signs that the time to start formulating recommendations has come. Alternatively, international commissions could also be organized under the auspices of transnational organizations, and while we have spoken singularly of the need for a commission to eventually develop guidelines for SHEEFs, it may be that multiple commissions will be needed to translate the general principles embodied in the guidelines into regulations for SHEEFs: Indeed, this is highly likely given that regulations for embryo and stem cell research are already implemented variously and to some extent discordantly around the world as laws, funding limits, and institutional policy mandates (*Hyun et al., 2016*; *Pera et al., 2015*), and SHEEF regulations would need to be reconciled with these. It is our hope that the analysis and proposals we have presented in this article will encourage these and like-positioned bodies to discuss the ethical issues raised by SHEEFs, both individually and collectively, and also serve as an initial concept for addressing them that they can debate, develop, organize about, and ultimately undertake.

## Note added in proof

As this article went to press, a report appeared that illustrates the rapid progress being made towards developing methods for generating synthetic entities with embryo-like features from mouse stem cells with remarkable morphological and molecular similarity to non-synthetic mouse embryos (*Harrison et al., 2017*). In this study, microcolonies of mESC and mouse trophoblastic stem cells grown in 3D culture self-organized and developed in a highly parallel manner to non-synthetic mouse embryos up through E6.5, an early post-implantation stage that includes initial phases of mesoderm generation.

## Acknowledgements

We very gratefully thank George Q Daley and M William Lensch for their extensive help and commentary, and the members of the Harvard ESCRO committee for other comments, on earlier versions of this manuscript, with particular thanks to ESCRO members Robert D Truog, Dan I Wikler, and I Glenn Cohen. We give further thanks to the Harvard ESCRO and the many participants and working groups in the symposium "The Ethics of Early Embryo Research & the Future of the 14-Day Rule" on 7-8 November 2016, who gave insightful comments on ideas

presented in this article, and we are particularly indebted to Melissa J Lopes (who organized) and Robert D Truog (who chaired) this symposium, which was co-sponsored by The Petrie-Flom Center, the Edmond J Safra Center for Ethics, the Harvard Medical School Center for Bioethics, and the Harvard Stem Cell Institute. We thank Je Hyuk Lee, Catherine Spina, James J Collins, and Henry T Greely for their work on the ESATE method and our 2011 submission to ESCRO. Finally, we give deep thanks to Janet Rossant and Sarah Chan for their very detailed and invaluable reviewers' comments on the version of the manuscript initially submitted to eLife. The content is solely the responsibility of the authors and does not necessarily represent the official views of the National Institutes of Health, the European Commission, nor the views of the individuals we acknowledge above. The depiction of the female reproductive organs in *Figure 1* was taken from the figure "Human Fertilization" by Ttrue12 under license CC-BY-SA-3.0.

John Aach is in the Department of Genetics, Harvard Medical School, Boston, United States

http://orcid.org/0000-0001-7111-9541

Jeantine Lunshof is in the Department of Genetics, Harvard Medical School, Boston, United States, and the Department of Genetics, University Medical Center Groningen, University of Groningen, Groningen, The Netherlands

http://orcid.org/0000-0002-5630-7947

Eswar Iyer is in the Department of Genetics, Harvard Medical School, Boston, United States

George M Church is in the Department of Genetics, Harvard Medical School, Boston, United States

*Author contributions:* JA, Conceptualization, Supervision, Visualization, Writing—original draft, Writing—review and editing; JL, Conceptualization, Supervision, Writing—review and editing; EI, Investigation, Writing—review and editing; GMC, Conceptualization, Supervision, Funding acquisition, Writing—review and editing

*Competing interests:* GMC: George Church is actively involved with companies commercializing synthetic biology. His financial interests are specified on his website: http://arep.med.harvard.edu/gmc/tech.html. Additionally, this article was supported by two grants from the NIH (Grants RM1HG008525 and P50HG005550). George Church is a founder and has a financial interest in two companies, Editas and ReadCoor, that are related to those grants. Additionally, potentially relevant patents to this article have been submitted. The companies have not had a role in the study design, data collection and analysis, decision to publish or preparation of the present manuscript. The other authors declare that no competing interests exist.

*Ethics:* No experiments conducted in connection with this study involved human subjects. On two occasions described in the text, experiments involving the use of hiPSC were brought before the Harvard Embryonic Stem Cell Research Oversight (ESCRO) committee and judged to be in accordance with Harvard ESCRO policies. The mouse embryos used in our 2011 experiments on Embryo Scaffold-Aided Tissue Engineering (ESATE) were obtained from Jackson Labs as fixed, frozen 15.5 dpc embryos derived from wild type (C57BL/6) mice, all of which were maintained and harvested according to their animal care protocols.

### Funding

| Funder | Grant reference number | Author |
| --- | --- | --- |
| National Human Genome Research Institute | RM1HG008525 | George M Church |
| National Human Genome Research Institute | P50HG005550 | George M Church |
| European Union's Seventh Framework Programme | 298698 | Jeantine Lunshof |

The funders had no role in study design, data collection and interpretation, or the decision to submit the work for publication.

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
