## [Decision Letter]

Thank you for submitting your article "Addressing the ethical issues raised by synthetic biology-generated organized human entities" to *eLife* for consideration as a Feature Article. Your article has been reviewed by two peer reviewers, and the evaluation has been overseen by Fiona Watt as the Reviewing and Senior Editor.

The reviewers have discussed the reviews with one another and the Reviewing Editor has drafted this decision to help you prepare a revised submission.

Summary:

Both reviewers feel that following revision the paper is likely to be a valuable contribution.

Essential revisions:

Please shorten the text and provide more specifics about how the oversight process would work. Also, *eLife* does not permit supplemental information, so please integrate the text and references in the supplemental material into the main text.

Reviewer #1:

In this commentary, the authors discuss the advances in the development of what they call synthetic human embryo-like entities (SHELEs) from human pluripotent stem cells and other sources. They focus on 'gastruloids' from human ES cells but also consider reconstituted decellularized tissues and fetuses, organoids and linked organoids or linked organs-on-a-chip. Given that we already know how to generate all three germ layers separately from hES cells in culture, I would also throw 3D tissue recombinants derived from stem cells but designed to establish anterior-posterior, dorsal-ventral and/or left-right patterning of the early body axis into the mix. They point out that these kinds of emergent properties bypass the normal events of early development and gastrulation and so are not easily encompassed by the 14-day rule on the ethical bounds of human embryo culture. This has been pointed out previously by others, probably most clearly by Pera et al. (2015).

To address these issues they propose the formation of a major new Embryo Research Ethics Guidelines Determination Process, whose focus would be to completely revisit the issue of the 'moral status' of the embryo in the light of new technologies and to propose working guidelines for the determination of what is acceptable and not acceptable research with SHELEs. It is not clear under what jurisdiction this commission would be set up, nor how its findings would be interpreted under different legal frameworks already in place worldwide to handle embryo research. The ISSCR revised stem cell guidelines just released in 2016, propose institutional human embryo research committees whose role would be to deal with all research involving entities that 'might manifest human organismal potential'. Is the idea that these committees would be informed by the overall guidance of the conclusions of the EREGDP? Would the HFEA be likely to be governed by the findings of the EREGDP?

In general, the issues raised here are interesting and important, but the article and its associated supplemental material are overly long, discursive and repetitive. By focusing on specific, explicit recommendations as to the formation and function of a proposed EREGDP, I think the authors miss the opportunity to inform a broad audience of the scientific and ethical dilemmas that are arising and will arise in future with the ongoing development of 'synthetic embryos'. I would propose that they merge the supplemental text into the main text, reduce redundancies and focus the article on discussion of their "Principle 1", rather than on the call for an EREGDP. At the end of the article they could suggest that the complexities of the issues might benefit from the formation of a new international commission to examine the issues.

1) Principle 1. Proposes to explore the 'moral status' of SHELEs. This is really the crux of the matter and I would suggest that this become the main focus of the article. They propose that it is necessary to identify those features of experimental systems that could signify that the 'moral status' is too high to proceed. I would caution strongly against the use of the term 'moral status' as this has been a highly contentious issue since the very beginning of research on human embryos. The fact is that no one agrees on when an embryo has 'moral status' – religious, societal and political viewpoints all play into this value-driven term. I do not think it would be possible to come to any consensus if moral status is to be the defining property. In other debates on this or similar issues with regard to human embryo research, interspecies chimeras, organoids, etc., the concepts of 'human organismic potential' and/or 'human defining features' are used to help inform the debate. I would favour this approach. In fact, in the supplemental data, the authors venture into what human defining properties of SHELEs might be of concern, e.g. acquisition of sentience and pain, fetal heartbeat, human form, twinning potential etc. I would take this section from supplemental and bring it into the main Discussion. This deals with 'human defining features' but does not deal with 'human organismic potential'. By the simplest definition of being able to generate a complete viable human being, all current SHELEs would fail this test. However, as the authors point out, generation of organs in vitro is not inconceivable and already organoids from liver and intestine are being successfully reintroduced in vivo. At what point does the development of a brain organoid in vitro transition towards development of a structure with potential to drive human personality if transplanted in vivo? This also needs to be considered in the Discussion.

2) Principle 2. Is really directed at the working of a future commission and suggests it needs to keep revisiting its guidelines. I think this is an unnecessary 'principle'. All good guidelines in this area have been and will continue to be updated at intervals, e.g. the NAS and ISSCR guidelines and the HFEA/ UK laws.

3) Principle 3 again is not really a new high level principle. It suggests that there should be some consideration of the scope and extent of embryo use. This is already in place in every jurisdiction that allows any research with human embryos – the justification for the research, the limits on the numbers of embryos needed and the process of informed consent are already part of every approval process (e.g. the recent Niakan and Lanner protocols).

4) Principle 4 suggests considering ways to limit the 'human potential' of SHELEs), perhaps revisiting the ideas behind the 'altered nuclear transfer' embryos proposed to limit the potential of somatic cell nuclear transfer embryos. There was considerable concern in the cloning debate that the ANT was essentially trying to bypass the fundamental ethical concerns and would not prevent the development of the cloning technology with its ultimate potential to clone humans. In the same way here, limiting some aspect of the future human potential of the SHELEs might permit researchers to go further down the path of human development in vitro than might be otherwise acceptable.

5) Reduce overall length by probably 50%. Focus on the uncertain ethical boundaries of this research and suggest what new questions need to be addressed to define those boundaries.

Reviewer #2:

Thank you for inviting me to review this interesting paper on a timely topic of high significance. The paper raises several important points; I feel that the explanation of the issue and the development of the argument could be refined in places, particularly with respect to discussions of moral status and the role of regulation. In particular, while I agree that the new lines of research described in the paper do require us to think further about how we regulate these areas of research, I am far from certain that the role of the review process proposed should be to make a determination on the biological features relevant to moral status that would then be used directly as a basis for developing regulation. I think any moves to review regulation or create new regulation in this area need to approach the issues much more broadly, rather than flattening them into a single moral problem with a scientific solution.

The authors are, I think, correct to suggest that gastruloids and SHELEs may challenge some of the assumptions that underpin existing regulatory approaches to human embryos; and that given the ethical concerns this may raise, we ought to consider how research on such entities should be regulated. Their criticism of the extent to which current regulation is tied to particular methods and materials is also well-directed; this has been a recurring problem throughout the history of regulating of embryo research.

Some of the analysis feels a little blunt, in that the authors often seem to take for granted that regulatory thresholds are intended directly to reflect moral status which in turn can be characterized purely in terms of physical phenomena. In fact, however, these are different types of concepts (legal / regulatory, ethical and scientific), and what the relationship between these concepts is within current regulatory frameworks, as well as what it should be in light of SHELEs, is a complex and not unproblematic question. Likewise the text seems in places to suggest that the current regulation is based on clear moral consensus, when in fact it is not; there continues to be deep disagreement over the moral status of the embryo, and while some accept the appearance of the PS either as a pre-emptive threshold (before which it is clear that no morally significant features have appeared) or a threshold of moral status in itself (after which something morally significant has changed, such as the emergence of a unique individual identity), the plurality of views within society about the moral status of the embryo is far broader. At other times the authors appear to espouse somewhat problematic ideas about morality, the role of regulation as well as what the objects of regulation are or should be. Morality does not easily translate directly into law, nor should we expect it to, and in no area is research with living beings currently regulated directly on the basis of moral status – compare for example regulatory frameworks for non-human animal research with those for research on human embryos or human beings.

I think this tendency to over-simplify the biology / morality / law relationship is partly a matter of framing and language, rather than the authors genuinely holding such views – they themselves note that there are diverse views about the moral status of embryos, and that the 14-day rule is the result of an "uneasy compromise". Nevertheless, it would be worth addressing these framing issues throughout the paper, in order to hone the argument.

I do think, however, that the authors are too quick to focus on moral status as the sole factor of importance in regulating embryo research, and thus I find it difficult to support the suggestion of an "Embryo Research Ethics Guidelines Determination Process" that is so heavily focused on determining moral status on the basis of biology.

---

## [Author Response]

Essential revisions:

Please shorten the text and provide more specifics about how the oversight process would work. Also, eLife does not permit supplemental information, so please integrate the text and references in the supplemental material into the main text.

Reviewer #1:

In this commentary, the authors discuss the advances in the development of what they call synthetic human embryo-like entities (SHELEs) from human pluripotent stem cells and other sources. They focus on 'gastruloids' from human ES cells but also consider reconstituted decellularized tissues and fetuses, organoids and linked organoids or linked organs-on-a-chip. Given that we already know how to generate all three germ layers separately from hES cells in culture, I would also throw 3D tissue recombinants derived from stem cells but designed to establish anterior-posterior, dorsal-ventral and/or left-right patterning of the early body axis into the mix. They point out that these kinds of emergent properties bypass the normal events of early development and gastrulation and so are not easily encompassed by the 14-day rule on the ethical bounds of human embryo culture. This has been pointed out previously by others, probably most clearly by Pera et al. (2015).

Section 1 now expands on the ways in which these entities may be created and briefly discusses 3D culture methods (first paragraph), and the idea of 3D printing of signaling centers along with the 3 germ layers is brought up in section 2 (second paragraph). We very much appreciate, and multiple times cite, Pera et al. (2015) for their excellent discussions of how the entities generated by Warmflash et al. and others recapitulate aspects of embryo development but are very unlike embryos, and of how these developments will increasingly impact embryo and stem cell regulations. But Pera et al. do not go into the points we make about how new stem cell and tissue engineering techniques may come to enable direct synthesis of embryo-like structures that allow stages like the PS to be specifically avoided or bypassed, and the specific implications of these methods for guidelines, and these are our focus.

To address these issues they propose the formation of a major new Embryo Research Ethics Guidelines Determination Process, whose focus would be to completely revisit the issue of the 'moral status' of the embryo in the light of new technologies and to propose working guidelines for the determination of what is acceptable and not acceptable research with SHELEs. It is not clear under what jurisdiction this commission would be set up, nor how its findings would be interpreted under different legal frameworks already in place worldwide to handle embryo research. The ISSCR revised stem cell guidelines just released in 2016, propose institutional human embryo research committees whose role would be to deal with all research involving entities that 'might manifest human organismal potential'. Is the idea that these committees would be informed by the overall guidance of the conclusions of the EREGDP? Would the HFEA be likely to be governed by the findings of the EREGDP?

A common element of feedback we heard from both reviewers and others who have given us comments has been that our proposed processes made too tight a connection between the exploratory inquiries into the conceptual/moral issues raised by SHELEs (now SHEEFs) and the determination of guidelines for regulating SHEEF experiments. Thus, we have completely rethought the way these two activities should be connected and have put much more room between the inquiries and the process that might eventually develop guidelines. The new manuscript focuses on and calls for the inquiries, but no longer makes any specific recommendation for an “EREGDP” (i.e., a “guidelines determinationprocess”) at all. Regarding the former, we now describe the conceptual and moral problems raised by SHEEFs and call for exploratory ethical and conceptual inquiries led by the research and bioethics communities along the lines we laid out in our original manuscript (now in sections 4.1-4.3 and 5). But instead of having all this proceed within the context of an EREGDP, we talk about how a commission may ultimately be desirable and necessary to formulate guidelines for SHEEF research, after the inquiries have clarified the conceptual/moral issues sufficiently to provide well-formulated alternatives to work with. Of the eventual commission, we say that it would have the primary task of deliberating on this input and coming to a collective judgment on the research limits that should apply to SHEEFs. These points are mainly elaborated in our central section 4.4 that describes our approach to what we still propose to be the core problem for determining research limits for SHEEFs: how to base these as directly as possible on conditions under which they may be ascribed moral status.

We come back to the question of who, specifically, in the bioethics and research communities, might organize these inquiries, and where the follow-on commission might eventually come from, in our Discussion section 6 at the end of the manuscript. Here we describe how ESCROs and ISSCR’s EMROs, as well as HFEA and similar authorities, are “well positioned” to take leadership roles, since these bodies already contain both researchers and bioethicists who are overseeing stem cell and embryo research, and, indeed, may already be reviewing research proposals involving SHEEF-like entities. We do not say how these ESCROs and EMROs (etc.) should organize or select a leadership group – we don’t think we, as researchers, have the expertise or authority to do this: Rather, we see ourselves as having identified a problem that would benefit from their attention and provided an initial list of concrete proposals and questions that need to be explored. (What we *can* mention here that our own Harvard ESCRO seems motivated to pursue these explorations: They already recently organized a symposium at which some of these matters were discussed in presentations and working groups.) We also mention ISSCR as a body that could not only play a role in organizing the inquiries, but that could also be “well positioned” to act as a commission that might eventually develop guidelines for SHEEF research. This is based on our perception that ISSCR *already is* a commission that has come to collective judgments regarding research (as well as clinical and translational) guidelines, and that it *already has* effectively issued the first guidelines for what we are calling “SHEEFs”. But, again, we don’t feel we are in a position to tell ISSCR what they should be doing – we can only suggest.

Finally, regarding the ISSCR’s recommendation regarding entities that “might manifest human organismal potential”, which we consider the first recommended guideline for SHEEFs, we briefly discuss this in the Introduction (third paragraph) and at the end of section 3, but have more to say on this in a response to another comment below. We think ISSCR should get tremendous credit for having recognized that these entities raise new ethical questions, and in the passage at the end of section 3 we note that the *way* ISSCR framed this guideline actually solves a structural problem we describe earlier in section 3. But there we also raise the question of whether this will work for *PS-bypassing* SHEEFs, since as written it requires that embryos and entities with “human organismal potential” be subject to the *same* rules. Indeed, in another rethinking of our original manuscript – triggered, again by the common advice from reviewers and others – we now make clear in section 5 (second paragraph) that embryos and SHEEFs *need not* end up with the same research limits. (Thus, in addition to its being a “guidelines determination process”, another problem with our original EREGDP conception was that it was an “*embryo* research” process that would seek common guidelines for embryo as well as SHEEF research!)

In general, the issues raised here are interesting and important, but the article and its associated supplemental material are overly long, discursive and repetitive. By focusing on specific, explicit recommendations as to the formation and function of a proposed EREGDP, I think the authors miss the opportunity to inform a broad audience of the scientific and ethical dilemmas that are arising and will arise in future with the ongoing development of 'synthetic embryos'. I would propose that they merge the supplemental text into the main text, reduce redundancies and focus the article on discussion of their "Principle 1", rather than on the call for an EREGDP. At the end of the article they could suggest that the complexities of the issues might benefit from the formation of a new international commission to examine the issues.

We thank reviewer 1 for saying that the issues we raise are “interesting and important,” and for the very highly insightful comments and criticisms she has made regarding the manuscript. In the new version of the manuscript, we’ve tried to change the focus and structure of the article exactly as suggested – and, (thanks again!!) we think this has make it quite a bit clearer! Specifically, we completely eliminated the Principles and have focused on the core ethical and conceptual issues raised by SHEEFs. We have also merged those parts of the old Supplemental Information document that we felt were germane and eliminated the rest, and tried hard to eliminate redundancies. The article remains long, but we think it is much improved and better focused, and that it will hold readers’ interest.

1) Principle 1. Proposes to explore the 'moral status' of SHELEs. This is really the crux of the matter and I would suggest that this become the main focus of the article. They propose that it is necessary to identify those features of experimental systems that could signify that the 'moral status' is too high to proceed. I would caution strongly against the use of the term 'moral status' as this has been a highly contentious issue since the very beginning of research on human embryos. The fact is that no one agrees on when an embryo has 'moral status' – religious, societal and political viewpoints all play into this value-driven term. I do not think it would be possible to come to any consensus if moral status is to be the defining property. In other debates on this or similar issues with regard to human embryo research, interspecies chimeras, organoids, etc., the concepts of 'human organismic potential' and/or 'human defining features' are used to help inform the debate. I would favour this approach. In fact, in the supplemental data, the authors venture into what human defining properties of SHELEs might be of concern, e.g. acquisition of sentience and pain, fetal heartbeat, human form, twinning potential etc. I would take this section from supplemental and bring it into the main Discussion. This deals with 'human defining features' but does not deal with 'human organismic potential'. By the simplest definition of being able to generate a complete viable human being, all current SHELEs would fail this test. However, as the authors point out, generation of organs in vitro is not inconceivable and already organoids from liver and intestine are being successfully reintroduced in vivo. At what point does the development of a brain organoid in vitro transition towards development of a structure with potential to drive human personality if transplanted in vivo? This also needs to be considered in the Discussion.

As noted above, we have closely followed the reviewer’s recommendations concerning the structure and focus of the article, and this has greatly improved the manuscript. The proposal covered by the old Principle 1 – the idea of trying to base research limits as directly as possible on conditions that indicate moral status – is now the main focus of the article and is discussed in depth in section 4, especially 4.1-3. The material that was in the Supplemental Information on other features such as heartbeat, human form, twinning potential, has been brought into this section. The material associated with other principles has either been eliminated or has been greatly reduced: What little is left is now part of section 5 (last two paragraphs).

However, we felt we could we could not follow some of the other recommendations offered in the above comment:

On the use of the term “moral status”: We understand and have considerable sympathy for the issue raised by the reviewer: Indeed, in the original drafts of the manuscript, we actually did try very hard to avoid this and related terms for the reasons you suggest! However, feedback we received from George Daley and Willie Lensch argued against this strongly, and, persuaded by their arguments, we ultimately wrote “moral status” back into the manuscript. We take the liberty of quoting an excerpt from an email George sent us on this matter: “I think your attempt to avoid fuzzy terms like “developmental potential” and “moral status” means you take yourself out of the debate rather than engage in it. The possibility that in vitro-derived embryo-like structures might approach equivalence to a native embryo is precisely the concern that drives the debate. If you wish to allay those concerns, then you must argue forcefully that whatever synthetic biology is creating in a dish is distinct from a native embryo in developmental potential and thus can safely be granted a distinct moral status.” Thus, much as we recognize the problems associated with the terminology of “moral status” raised by the reviewer, we feel we must use it despite its deficiencies.

The question of “human defining features” vs. “human organismic potential” vs. “moral status” raises more complex and interesting issues that we’d love to debate at length with the reviewer, but in these responses we will limit ourselves to saying that we don’t see these other terms as offering much in the way of clarity over “moral status”. For instance, consider the sense of the term “human” in these expressions: Given that the features and capacities involved – “fetal heartbeat”, “twinning potential”, and “sentience” – are found in non-human animal as well as human embryos, they cannot literally be “*human* defining”, so that the meaning that is conveyed by the term “human” in them can only be that they contribute to the human embryo’s becoming a “human person.” But to us, that adds no clarity over saying that they indicate that the embryo acquires greater moral status when they develop the feature, and that is essentially how we describe it. On a subtler level, if “human” in these expressions is effectively a synonym for “human person,” calling these features “human *defining*” seems to us to get the relationship wrong, as having a human fetal heartbeat doesn’t seem to be a “defining” feature of a “human person.” The right relationship, it seems to us, is that it is a feature that is taken to *signify*, or that is *associated* with and used as a basis for attributing, an increase in moral status in the embryo: hence our terminology “moral status signifying feature”. Finally, (and as perGeorge Daley), using “moral status” terminology here connects up directly with previous treatments of the subject. Indeed, our discussion of these features comes right out of 1994 NIH Report of the Human Embryo Research Panel, which clearly and unhesitantly identifies these features as having “moral significance” and as indicators of “moral status.”

The concept of “human organismal potential” raises some other considerations. It homes in on the *developmental potential* that an entity has to become a “human organism” – where, again, the salient sense of this expression seems to be “a developing organism that is developing into a human person.” So, this term again seems to us to covertly derive its meaning from traditional notions of moral status, with focus on the complex problem of how *developmental potential* figures into determination of such status. Here we have two reactions: First, the good thing about this term is that it is wide enough to accommodate some entities that aren’t natural humans – which means it can cover at least some kinds of SHEEF! But the second is that, to our way of thinking, one can’t limit assessment of this potential to the ability to attain the ultimate endpoints of natural embryogenesis– i.e., one can’t limit consideration to the ability to “generate a complete viable human being”, or to a brain organoid that has the “potential to drive human personality if transplanted in vivo”. This is because even for non-synthetic human embryos, attributions of moral status don’t seem to depend on the embryo being able to make it to fully viable human being: Indeed, the 14-day rule itself seems to exemplify this. The rule helps avoid a condition that would be morally concerning were it to occur – the development of an embryo that could come to sentience and experience pain – but this rule is applied even though *other* rules would prohibit such an embryo from ever coming to full viability, since embryos used in experiments are not allowed to be implanted. Thus, embryos in experiments are still accorded enough moral status to protect them from exposure to the possibility of pain, even though they would not ever be allowed to develop fully. (Louis Guenin makes similar points in his “Morality of Embryo Use,” chapter 8.) Our contention is that if non-synthetic embryos can have moral status even without full developmental potential, the situation for SHEEFs needs to be closely examined, too. While it is not presently foreseeable that SHEEFs could be brought to a point of full viability, what seems quite possible is for SHEEFs to have (e.g.) the neural circuitry to experience pain, and to be kept alive and in this condition indefinitely by virtue of continuous life support (through perfusion apparatus).

This comes back to what we *like* about the ISSCR’s and Reviewer’s use of the term “human organismal potential.” As the first guideline that is framed to accommodate the possibility of SHEEFs, it effectively makes ‘room at the table’ for SHEEFs as kinds of entity that may need to be incorporated into ethics-based research guidelines – and in our view, ISSCR gets tremendous credit for this move! But the vague phrase “human organismal potential” is effectively only a ‘placesetting’ at the table, and leaves very much open the question of who will actually be invited to sit there. Our view is that the inquiries we have proposed need to be done to figure that out!

2) Principle 2. Is really directed at the working of a future commission and suggests it needs to keep revisiting its guidelines. I think this is an unnecessary 'principle'. All good guidelines in this area have been and will continue to be updated at intervals, e.g. the NAS and ISSCR guidelines and the HFEA/ UK laws.

The manuscript is no longer organized around the “Principles,” and there is no longer anything said in the manuscript about the need for a recurring process.

3) Principle 3 again is not really a new high level principle. It suggests that there should be some consideration of the scope and extent of embryo use. This is already in place in every jurisdiction that allows any research with human embryos – the justification for the research, the limits on the numbers of embryos needed and the process of informed consent are already part of every approval process (e.g. the recent Niakan and Lanner protocols).

4) Principle 4 suggests considering ways to limit the 'human potential' of SHELEs), perhaps revisiting the ideas behind the 'altered nuclear transfer' embryos proposed to limit the potential of somatic cell nuclear transfer embryos. There was considerable concern in the cloning debate that the ANT was essentially trying to bypass the fundamental ethical concerns and would not prevent the development of the cloning technology with its ultimate potential to clone humans. In the same way here, limiting some aspect of the future human potential of the SHELEs might permit researchers to go further down the path of human development in vitro than might be otherwise acceptable.

Again, the “Principles” are gone, but very brief discussions of the potential impact of SHEEFs on embryo experiments, and of the possibility of using an ANT-like strategy for generating SHEEFs that can model embryogenesis closely, are given in section 5 (third paragraph).

5) Reduce overall length by probably 50%. Focus on the uncertain ethical boundaries of this research and suggest what new questions need to be addressed to define those boundaries.

We did what we could to follow these suggestions and this resulted in very substantial reductions in text: For instance, the ~8 pages (166 lines) of text that formerly dealt with Principles 3 and 4 were entirely eliminated and replaced in section 5 (third paragraph, 23 lines) with ~1 page that very briefly summarize the issues they covered. However, reductions in length from cutting out the Principles and extraneous discussion were offset by inclusions of material from the old Supplemental Information (SI) document (see below), and amplifications related to points raised by the reviewers and others. As a result, the manuscript is actually ~24 lines (~1 page) longer than it originally was. But we hope the reviewer agrees with us that the result is much clearer and better focused than the original.

Reviewer #2:

Thank you for inviting me to review this interesting paper on a timely topic of high significance. The paper raises several important points; I feel that the explanation of the issue and the development of the argument could be refined in places, particularly with respect to discussions of moral status and the role of regulation. In particular, while I agree that the new lines of research described in the paper do require us to think further about how we regulate these areas of research, I am far from certain that the role of the review process proposed should be to make a determination on the biological features relevant to moral status that would then be used directly as a basis for developing regulation. I think any moves to review regulation or create new regulation in this area need to approach the issues much more broadly, rather than flattening them into a single moral problem with a scientific solution.

We very much appreciate the reviewer’s description of our manuscript as an “interesting paper on a timely topic of high significance”! Reviewer 1 and others raised similar issues. These comments helped us see that our proposed processes made too tight a connection between the exploratory inquiries into the conceptual/moral issues raised by SHELEs (now SHEEFs), and what might have to be done to turn the results of these inquiries into guidelines that would apply to SHEEF experiments.

In the new and completely rewritten manuscript, we focus virtually entirely on the former and only talk about the latter as an eventual follow-on process that would require a “commission”. The manuscript no longer refers to an “EREGDP” (i.e., a “guidelines determinationprocess”) at all, and all talk about the “Principles” that such a process might follow has been eliminated. In describing the conceptual and moral problems raised by SHEEFs, and calling for exploratory ethical and conceptual inquiries into these, we still focus on the need to identify the biological substrates of features relevant to moral status. But the results of these inquiries don’t lead to guidelines or regulation in any direct way. We suggest that the inquiries be led by the research and bioethics communities, and that they solicit input and advice from other (external) communities, institutions, and traditions, along the lines laid out in the first version of our manuscript. We view the results of these inquiries as providing information, increased conceptual clarity, and systematic thinking about the moral concerns raised by SHEEFs and the biology associated with these moral concerns. The concepts and systematic thinking that result might not all agree, and, indeed, might be expected to give rise to disagreeing viewpoints as to how to deal with SHEEFs. We say of the commission that it could come into being when the inquiries had produced these variously consolidated viewpoints, and, as a primary task, deliberate on these alternatives, come to “collective judgment” about them (more on this below), and fashion guidelines for those aspects that need them.

These points are elaborated in our core section 4. In our Discussion section 6, we provide a few ideas about what bodies might be positioned to lead the inquiries, and what other bodies might be positioned to be the eventual commission: Specifically, we suggest that ESCRO and EMRO committees might be well positioned to lead the inquiries, as they are already bring together scientific researchers, bioethicists, and ‘outside’ members who are overseeing embryo and stem cell research. In some cases these committees might already be overseeing some SHEEF research. We think bodies like ISSCR might be well positioned to eventually become commissions that develop guidelines for SHEEF research – and, indeed, ISSCR has already recommended a research limit regarding entities with “human organismal potential” that relates to SHEEFs. However, these suggestions are offered only as starting points for discussion, as we feel that we have neither the expertise nor authority to instruct these bodies on whether and how they might self-organize on these matters.

As we said above, we made these changes as a way of addressing what we perceived as a common criticism received not only from reviewer 2, but also from reviewer 1 and other sources. But reviewer 2 put strong emphasis on “regulation” in a way that was not brought up by other feedback. While we respond to specific comments on this below, we preface these with a high-level response here:

We feel that reviewer has rightly pointed out that our original manuscript did not deal with the fact that embryo and stem cell research is subject to a wide variety of legal, funding-level, and institutional policy controls, and also rightly noted that while these may be motivated by moral concerns, they do not reflect these concerns in any direct or simple way. However, it was never our intention to suggest that our proposed processes would systematically address these various forms of control, and the same goes for the revised set of processes outlined in our new manuscript. We note that where reviewer 2 generally talks about “regulations”, we have always talked about “guidelines.” On our reading, the term “regulations” covers a detailed realm of specific obligations and controls governing behavior, funding, and sanctions encoded in law and institutional policy, while “guidelines” are broader instructions regarding what should or shouldn’t be done that do not necessarily specify detailed forms of control.

These usages conform to dictionary definitions: “guideline” = “an indication or outline of policy or conduct” (https://www.merriam-webster.com/dictionary/guideline), while “regulation” = “an authoritative rule dealing with details or procedure” or “a rule or order issued by an executive authority or regulatory agency of a government and having the force of law” (https://www.merriam- webster.com/dictionary/regulation).

But reviewer 2’s comments made us reflect on why our call to develop “guidelines” might be read as a call for systematic review / revision of “regulations”, and we believe we are now more sensitized and educated to the issues. A contributing factor may have been that here in the US where we are located, the main thrust of federal policies on embryo and stem cell research has been to demarcate ethically sensitive areas that should not be federally funded, so that it has been left up to non-governmental bodies like the US National Academy of Sciences to formulate detailed “guidelines” about how such research should be conducted that are not rendered into federal law (although the situation may be different at the State level). The situation is, of course, very different elsewhere in the world, as documented by Pera et al. and illustrated historically by the Warnock commission, which *did* recommend changes to law (e.g., in Section 11.16). These reflections made us realize that our call for “guidelines” could be interpreted as a call for “regulations” as a matter of course for researchers and bioethicists who work in or with non-US jurisdictions. We thank reviewer 2 for helping us to see this!

Nevertheless, our focus remains “guidelines,” and thus our task is to make this clear. We hope the following features of the new manuscript work towards this goal: (1) First, the change in focus described above, whereby our principal recommendation is for a set of exploratory inquiries into ethical issues surrounding SHEEFs vs. a follow-on commission for determining guidelines, should go a long way towards making it clear that we are not calling for a systematic review/revision of regulations. (2) We now state more explicitly that the focus of the article is on the scientific and ethical underpinnings of research limits like the 14-day rule (emphasis added):

“The 14-day rule arose from recommendations by a series of commissions dating back to 1979 (Warnock, 1984; Munthe, 2001; Takahashi et al., 2007) charged with working out ethics-based guidelines for embryo creation and usage, initially focused on assisted reproductive technologies (ART) but later extended to stem cells and other biological constructs. […] Here we call these latter “research limits” to distinguish them from these other kinds of guidelines, and note that we not only focus exclusively on such limits vs. other kinds of guidelines, but also on the ethical and scientific reasons for establishing these limits vs. the ways in which they may be variously rendered as laws, policies, or other regulations.”

Finally, (3) we come back to the difference between “guidelines” and “regulations” at the end of section 6 Discussion in the context of the work of eventual commissions:

“…while we have spoken singularly of the need for *a* commission to eventually develop guidelines for SHEEFs, it may be that multiple commissions will be needed to translate the general principles embodied in the guidelines into regulations for SHEEFs: Indeed, this is highly likely given that regulations for embryo and stem cell research are already implemented variously and to some extent discordantly around the world as laws, funding limits, and institutional policy mandates (International Society for Stem Cell Research. 2016; Oye et al., 2014), and SHEEF regulations would need to be reconciled with these.”

We hope the statements in (2) and (3), combined with our general change in focus (1), will be enough to distinguish “guidelines” from “regulation” and establish that our primary direction is for the former. We considered various ways of defining and distinguishing “guidelines” vs. “regulation” explicitly at the beginning of the paper, but all attempts seemed to us to detract from the flow and organization of our main arguments and proposals about the ethical problems raised by SHEEFs – and these latter are our top priority! Beyond these steps, we continue to refer to “guidelines” throughout to avoid encouraging confusion with “regulation”, except in a few cases where reviewer 2’s comments have made us see that the latter term is more accurate.

The authors are, I think, correct to suggest that gastruloids and SHELEs may challenge some of the assumptions that underpin existing regulatory approaches to human embryos; and that given the ethical concerns this may raise, we ought to consider how research on such entities should be regulated. Their criticism of the extent to which current regulation is tied to particular methods and materials is also well-directed; this has been a recurring problem throughout the history of regulating of embryo research.

We thank the reviewer for agreeing with us on these key points.

Some of the analysis feels a little blunt, in that the authors often seem to take for granted that regulatory thresholds are intended directly to reflect moral status which in turn can be characterized purely in terms of physical phenomena. In fact, however, these are different types of concepts (legal / regulatory, ethical and scientific), and what the relationship between these concepts is within current regulatory frameworks, as well as what it should be in light of SHELEs, is a complex and not unproblematic question. Likewise the text seems in places to suggest that the current regulation is based on clear moral consensus, when in fact it is not; there continues to be deep disagreement over the moral status of the embryo, and while some accept the appearance of the PS either as a pre-emptive threshold (before which it is clear that no morally significant features have appeared) or a threshold of moral status in itself (after which something morally significant has changed, such as the emergence of a unique individual identity), the plurality of views within society about the moral status of the embryo is far broader. At other times the authors appear to espouse somewhat problematic ideas about morality, the role of regulation as well as what the objects of regulation are or should be. Morality does not easily translate directly into law, nor should we expect it to, and in no area is research with living beings currently regulated directly on the basis of moral status – compare for example regulatory frameworks for non-human animal research with those for research on human embryos or human beings.

I think this tendency to over-simplify the biology / morality / law relationship is partly a matter of framing and language, rather than the authors genuinely holding such views – they themselves note that there are diverse views about the moral status of embryos, and that the 14-day rule is the result of an "uneasy compromise". Nevertheless, it would be worth addressing these framing issues throughout the paper, in order to hone the argument.

As noted above, our primary recommendations are now for a research and bioethics community-led effort to explore the scientific and ethical issues surrounding SHEEFs, and though we hope this eventually will lead to a commission, we see the commission as recommending “guidelines” vs. “regulations”. However, we discuss the reviewer’s remarks about role of the PS in current guidelines / regulations below in response to another comment.

On the issue of “moral consensus”, we were counting on the fact that the term “consensus” does not imply ‘unanimous acceptance’, our description of the 14-day rule as an “uneasy compromise,” and our explicit mentions of dissenting views regarding the 14-day rule and the question of embryo moral status generally, to make it clear that we were aware of the complexity of these issues. Nevertheless, the term “consensus” can rather vaguely suggest that ‘everyone agreed’, so we now use the terms “collective judgment” or “collective agreement” in places where we had originally used “consensus.” Thus, for instance, we say that the commissions that adopted the 14-day rule “came to a collective judgment” on the rule (e.g., Introduction, second paragraph), and similarly say that the possible eventual commission on SHEEFs will need to “collective agreement” on guidelines (e.g., section 4.4, first paragraph). These terms seem less prone to being interpreted as implying unanimity, and it seems clear that, whatever else the original commissions may have done to adopt the 14-day rule, they at least made “collective judgments” about it. In our view, it is unfortunate that these commissions weren’t more explicit about their group decision processes, as these would be useful to know! But it is interesting to note that in several places the Warnock report specifically talks about how a “majority” of the commission agreed to a recommendation, while the 1994 NIH Report of the Human Embryo Research Panel does not seem refer to ‘majority’ decisions in this way (so we suspect that they operated more informally).

I do think, however, that the authors are too quick to focus on moral status as the sole factor of importance in regulating embryo research, and thus I find it difficult to support the suggestion of an "Embryo Research Ethics Guidelines Determination Process" that is so heavily focused on determining moral status on the basis of biology.

The manuscript no longer anywhere refers to an “Embryo Research Ethics Guidelines Determination Process,” and we hope our focus on exploratory inquiries and the steps we took to distinguish “guidelines” from “regulations” will make it clear that we are not expecting the inquiries to translate in any direct manner into regulations. However, we do still think that the *central* ethical questions raised by SHEEFs are whether and when they can or should be considered to have moral status on the basis of their possession of features associated with such status in embryos, and these remain the core questions of our proposed inquiries.